# Genomic diversity of non-diarrheagenic fecal *Escherichia coli* from children in sub-Saharan Africa and south Asia and their relatedness to diarrheagenic *E. coli*

Tracy H. Hazen[1,2], Jane M. Michalski[1,2], Sharon M. Tennant[2,3,4] & David. A. Rasko[1,2] ✉

*Escherichia coli* is a frequent member of the healthy human gastrointestinal microbiota, as well as an important human pathogen. Previous studies have focused on the genomic diversity of the pathogenic *E. coli* and much remains unknown about the non-diarrheagenic *E. coli* residing in the human gut, particularly among young children in low and middle income countries. Also, gaining additional insight into non-diarrheagenic *E. coli* is important for understanding gut health as non-diarrheagenic *E. coli* can prevent infection by diarrheagenic bacteria. In this study we examine the genomic diversity of non-diarrheagenic fecal *E. coli* from male and female children with or without diarrhea from countries in sub-Saharan Africa and south Asia as part of the Global Enteric Multicenter Study (GEMS). We find that these *E. coli* exhibit considerable genetic diversity as they were identified in all *E. coli* phylogroups and an *Escherichia* cryptic clade. Although these fecal *E. coli* lack the characteristic virulence factors of diarrheagenic *E. coli* pathotypes, many exhibit remarkable genomic similarity to previously described diarrheagenic isolates with differences attributed to mobile elements. This raises an important question of whether these non-diarrheagenic fecal *E. coli* may have at one time possessed the mobile element-encoded virulence factors of diarrheagenic pathotypes or may have the potential to acquire these virulence factors.

*Escherichia coli* is a diverse species of enteric bacteria that can cause severe clinical outcomes including diarrhea, urinary tract infections, meningitis, and blood stream infections[1–3]. *E. coli* is also prevalent as a member of the healthy human gastrointestinal microbiota, as it was detected in 61% of the stool microbiomes from healthy individuals of the human microbiome project (HMP)[4], and in other studies has been estimated to occur in >90% of humans[5]. The presumably non-

diarrheagenic 'commensal' *E. coli* are known to more successfully compete for nutrients during colonization of the gastrointestinal tract[6] and may have a role in preventing the colonization and disease caused by diarrheagenic *E. coli* such as the O157:H7 enterohemorrhagic *E. coli* (EHEC)[7,8], as well as other species of diarrheagenic bacteria[9].

The majority of research conducted on *E. coli* has focused on the pathogenic types (pathotypes) of *E. coli*, which have been linked to a

[1]Institute for Genome Sciences, University of Maryland School of Medicine, Baltimore, MD 21201, USA. [2]Department of Microbiology and Immunology, University of Maryland School of Medicine, Baltimore, MD 21201, USA. [3]Center for Vaccine Development and Global Health, University of Maryland School of Medicine, Baltimore, MD 21201, USA. [4]Department of Medicine, University of Maryland School of Medicine, Baltimore, MD 21201, USA. ✉e-mail: drasko@som.umaryland.edu

wide array of human illness[1–3,10] and significant foodborne outbreaks[11–13]. Multiple diarrheagenic pathotypes of *E. coli* were identified as a leading cause of severe childhood diarrhea by the Global Enteric Multicenter Study (GEMS), a three-year case-control epidemiological investigation of the causes of diarrhea among children <5 years old in seven countries of south Asia and sub-Saharan Africa[14,15]. GEMS examined the disease association of viruses, parasites, and bacteria, which included the four of the most prevalent pathotypes of diarrheagenic *E. coli*: the enterotoxigenic *E. coli* (ETEC), enteropathogenic *E. coli* (EPEC), enterohemorrhagic *E. coli* (EHEC) or Shiga-toxigenic *E. coli* (STEC), and the enteroaggregative *E. coli* (EAEC). Among the *E. coli* examined in GEMS, the heat-stable toxin ETEC (ST-ETEC) and EPEC isolates were linked to an increased risk of death[15]. While diarrheagenic pathotype *E. coli* were isolated both from children with and without diarrhea in GEMS[15–17], the carriage of known pathotype-associated diarrheagenic virulence factors by these *E. coli* underscores their potential to cause diarrhea under the right host conditions.

Notably, an *E. coli* isolate was obtained from 96% of the >22,000 children enrolled in GEMS[15] and ~50% of these children had an *E. coli* that was not identified as one of the four examined diarrheagenic pathotypes, thus representing non-diarrheagenic *E. coli* (non-DEC). Several previous studies have examined the genomic diversity of ETEC or EPEC isolates from children with or without diarrhea in GEMS[16–19]; however, there have been limited efforts to examine the genomic diversity of non-diarrheagenic fecal *E. coli* from GEMS. Overall, non-diarrheagenic fecal *E. coli*, especially non-DEC fecal isolates from children in low and middle income countries (LMICs) of south Asia and sub-Saharan Africa, are greatly understudied compared to diarrheagenic pathotype *E. coli*[20–22]. Thus, to gain insight into the non-diarrheagenic fecal *E. coli* from young children in these regions we examined the genomic diversity of *E. coli* from GEMS that could not be classified to a diarrheagenic pathotype, hereafter referred to as 'GEMS non-DEC fecal isolates'.

In this work we demonstrate that GEMS non-DEC fecal isolates from children with and without diarrhea are genetically diverse, as they were identified in all *E. coli* phylogroups and an *Escherichia* cryptic clade. Although the GEMS non-DEC fecal isolates lack the characteristic virulence factors of the diarrheagenic pathotype *E. coli*, they exhibit remarkable genomic similarity to previously described diarrheagenic isolates, with differences attributed to mobile elements. We also demonstrate that acquired antibiotic resistance genes and plasmids are prevalent among these GEMS non-DEC fecal isolates, especially among isolates associated with diarrhea.

## Results

### *E. coli* lineages containing the GEMS non-DEC fecal isolates
Phylogenomic analysis demonstrated considerable diversity among the GEMS non-DEC fecal isolates, with isolates identified in all *E. coli* phylogroups (Fig. 1, Supplementary Fig. 1a, and Supplementary Data Set 1). There were 33% (98/294) of the GEMS *E. coli* in phylogroup A, followed by 22% (66/294) in B1, 19% (55/294) in D, 8% (23/294) in B2, and the remaining isolates in phylogroups C (4%, 12/294), E (2%, 5/294), F (4%, 13/294), G (1%, 3/294), and cryptic clade I of *Escherichia* (5%, 16/294)[23,24] (Fig. 1, Supplementary Fig. 1a, and Supplementary Data Set 1). These trends are consistent with previous studies that demonstrated phylogroup B2 *E. coli* are more prevalent in North America and Europe, whereas phylogroup A and B1 *E. coli* are more prevalent in Africa, Asia, and South America[5,22]. There were similar proportions of isolates from both cases and controls identified in each phylogroup, as well as in cryptic clade I (Fig. 1, Supplementary Fig. 1b, and Supplementary Data Set 2). Overall, the GEMS non-DEC fecal isolates from cases and controls were similarly distributed among the phylogroups with no phylogroup containing significantly more case or control isolates (Supplementary Data Set 2 and Supplementary Fig. 1b). Also, there

were similar distributions of phylogroups among cases and controls each at each GEMS geographic site; however, the overall prevalence of phylogroups differed when comparing between geographic sites. For example, India had no isolates of phylogroups C, D, or E, and there was a greater proportion of isolates identified in phylogroup F (Fig. 1c).

We further examined the diversity of the GEMS non-DEC fecal isolates by their multilocus sequence type (ST) lineages. There were 142 different STs identified among the 294 GEMS non-DEC fecal isolates with 91 STs represented by a single isolate. Also, there were 27 newly designated STs among the GEMS non-DEC, highlighting the diversity of these isolates (Supplementary Data Set 1). The GEMS non-DEC fecal isolates were also grouped into ST lineages of closely related isolates. The ST lineages had significant bootstrap support (≥95%) and contained from one to nine different STs. The 294 GEMS non-DEC and 154 reference *E. coli* that included diarrheagenic pathotypes (EPEC, ETEC, EAEC, STEC, DAEC, EIEC), extraintestinal pathogenic *E. coli* (ExPEC) and previously described 'commensal' *E. coli* and were identified in 71 ST lineages that contained two or more genomes (Fig. 1 and Supplementary Data Set 3). An additional 49 GEMS non-DEC isolates and the pathotype reference genomes were not closely related to any other genomes and represent singletons. The greatest number of SNP differences in a single lineage was identified for the genomes of ST70 of phylogroup D that were all the same ST and had between 0 and 9740 SNP differences (median: 449) (Supplementary Data Set 3).

The lineage that exhibited the greatest genetic diversity was ST10 encompassing nine distinct STs (6 SLVs, 2 DLVs), up to 9385 SNP differences (median: 3336) (Supplementary Data Set 3). This lineage contained 12 GEMS non-DEC isolates from cases and 14 from controls, as well as ETEC, EPEC, EAEC, DAEC, STEC, and 'commensal' reference genomes. There were 22 ST lineages that contained only the GEMS non-DEC fecal isolates, demonstrating there is considerable unexplored genetic diversity among the non-diarrheagenic fecal *E. coli*. GEMS non-DEC fecal isolates from cases and controls were identified in similar numbers of lineages, with case isolates identified in 45 lineages and isolates from controls in 43 lineages (Supplementary Data Set 3). Six lineages (ST452, ST12296, ST12300, ST2016, ST2178, ST542) contained exclusively GEMS non-DEC isolates from controls, while four lineages contained exclusively GEMS non-DEC isolates from cases (ST12, ST117, ST4388, and ST361) (Supplementary Data Set 3). Two of the lineages of GEMS non-DEC fecal isolates from controls represent newly designated STs (ST12300 and ST12296). Further investigation is necessary to determine whether these *E. coli* lineages may be predictive of gut health or lack thereof among children in these regions.

### Genomic similarity of the GEMS non-DEC fecal isolates to pathotype *E. coli*
The phylogenomic analysis and comparison of the core SNP differences demonstrated that many of the GEMS non-DEC fecal isolates exhibit considerable genomic similarity to pathotype reference isolates, including those linked to severe illness (Fig. 2 and Supplementary Fig. 2). There were 46% (137/294) of the GEMS non-DEC isolates identified in 30 lineages that contained one or more pathotype reference isolates. Of these were 14 lineages that contained ETEC and 28% (81/294) of the GEMS non-DEC fecal isolates (Supplementary Data Set 3). Also, there were 13% (37/294) of the GEMS *E. coli* identified in five lineages that contained EPEC references; however, 26 of the isolates were identified in ST10, which is a lineage that contains previously described EPEC, ETEC, EAEC, DAEC, UPEC, and 'commensal' *E. coli*. GEMS non-DEC fecal isolates were identified in the same lineages as notable diarrheagenic pathotype *E. coli* such as a GEMS ETEC isolate associated with severe diarrhea[17] (Fig. 3a), or the Shiga-toxigenic O104:H4 German outbreak isolate C227-11[25] (Fig. 3b, Supplementary Data Set 4, see Supplementary Information).

As the GEMS *E. coli* isolates were originally classified by PCR to only the four most prevalent diarrheagenic pathotypes (EAEC, EHEC/

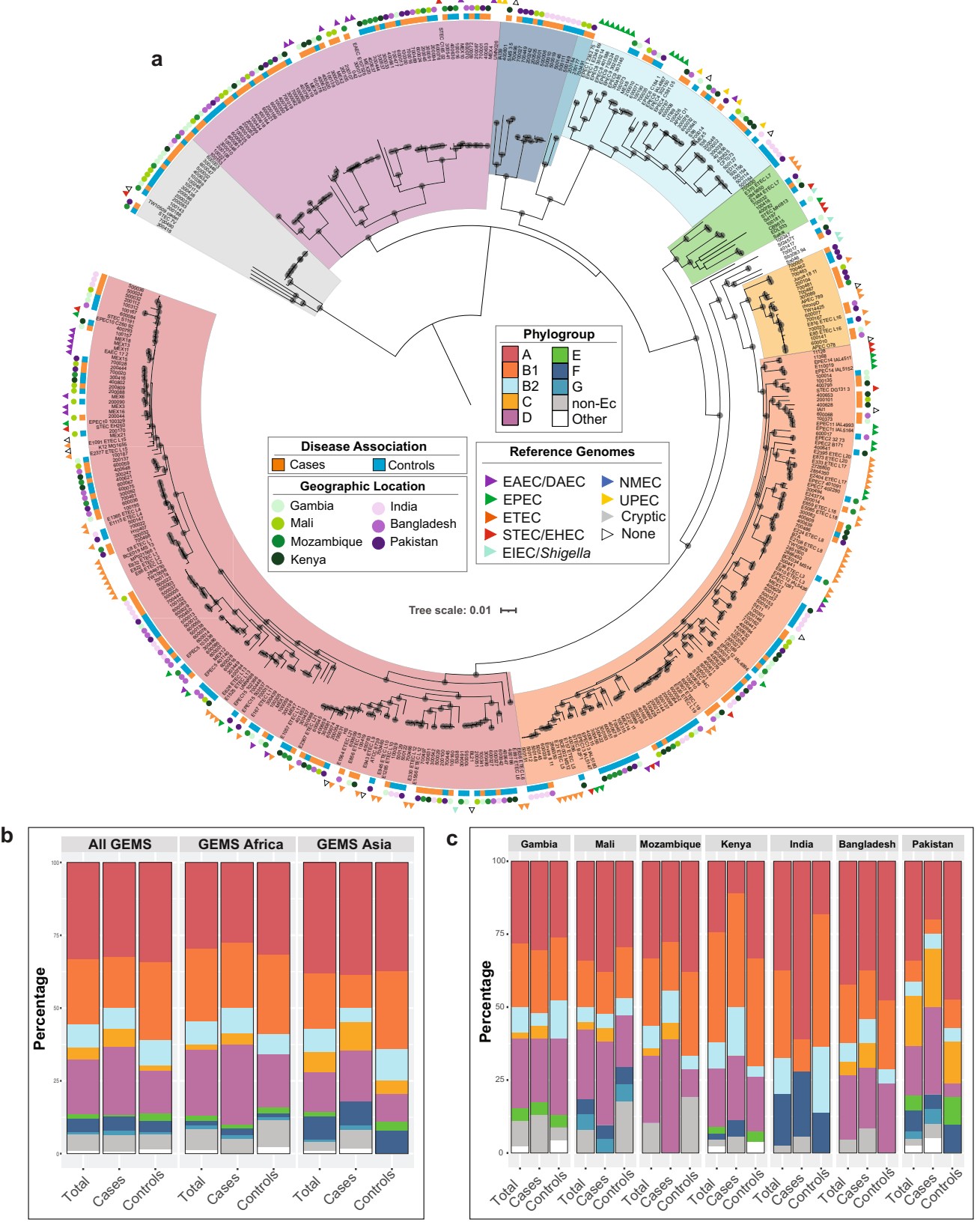

STEC, EPEC, or ETEC[26], it was not known whether some of these fecal *E. coli* may represent additional pathotypes, including diffusely adhering *E. coli* (DAEC) or enteroinvasive *E. coli* (EIEC), or the extra-intestinal pathogenic *E. coli* (ExPEC) such as uropathogenic *E. coli* (UPEC) and neonatal meningitis *E. coli* (NMEC)[1,10]. Here, we identified DAEC-associated adhesins (*afaE, draE,* and *daaE*)[27–29] in only 5% (15/

294) of the GEMS non-DEC fecal isolates (Supplementary Data Set 2). Additional genes of the Afa/Dr operon, which have also been identified in UPEC[29], were in 11% (33/294) of the GEMS non-DEC. This suggests DAEC may not be prevalent among children in south Asia and sub-Saharan Africa or at least not among GEMS enrollees, but would require further investigation to understand the prevalence of DAEC in

**Fig. 1 | Phylogenomic analysis of the GEMS non-DEC fecal isolates.**
**a** Phylogenomic analysis of the 294 GEMS non-DEC genomes and 154 previously characterized reference genomes representing the different *E. coli* pathotypes, *Shigella* species, and cryptic *Escherichia* lineage I (Supplementary Data Set 1). The *E. coli* phylogroups (A, B1, B2, C, D, E, F, and G) and *Escherichia* cryptic clade I are indicated by clade colors (see inset legend). Squares and circles next to each isolate label indicate the disease association (case vs. control) and geographic location (countries). Triangles indicate the *E. coli* pathotype or *Shigella* species of each reference genome while white ("None") indicates 'commensals' or other isolates that were not identified to one of the *E. coli* pathotypes. See inset figure legend for the specific color associations. **b** Percentage of GEMS non-DEC fecal isolates identified in each of the *E. coli* phylogroups, represented overall and by continent. **c** Percentage of GEMS non-DEC fecal isolates from different GEMS sites identified in each of the *E. coli* phylogroups. The inset legend indicates the color of each *E. coli* phylogroup, as well as cryptic lineage I (gray), and genomes that were outside of the designated phylogroups (white). The counts and percentages of GEMS non-DEC fecal isolates in each phylogroup are provided in Supplementary Data Set 2.

these regions. One or more virulence factors associated with NMEC, such as the cytotoxic necrotizing factor *cnf1*, S fimbriae, *ibeABC* involved in invasion, and *neuABCDE* of the K1 capsule[30] were also identified in only 5% (16/294) of the GEMS non-DEC isolates. The P fimbriae often associated with UPEC were identified among only 6% (18/294) of the GEMS isolates. There were 65 (22%) of the GEMS non-DEC fecal isolates identified in 16 of the top 20 ExPEC-containing lineages[31]; however, several of these lineages such as ST10 have also been described with diarrheagenic *E. coli* and are not exclusively ExPEC lineages. These findings demonstrate that other pathotypes such as DAEC, NMEC, and UPEC are not prevalent among this collection of GEMS non-DEC fecal isolates. Further studies would be necessary to determine whether these isolates, particularly those in lineages with previously described pathotype isolates, are able to cause illness in their current state or following the acquisition of some of the pathotype-specific virulence factors.

## Association of virulence factors with phylogroup, diarrhea, and geographic location

While the pathotype-specific virulence factors are central to the virulence mechanism of each pathotype[1], additional genetic loci have been described among *E. coli* that are involved in colonization and/or confer an advantage during pathogenesis[20,32,33]. However, much less is known about the role of virulence factors among non-diarrheagenic fecal *E. coli* that are not included in the pathotype designations. The most prevalent virulence factors identified among the GEMS non-DEC fecal isolates have roles in iron acquisition (enterobactin), adherence (*E. coli* common pilus, ECP)[32], or protein secretion (type II secretion, T2SS) (Fig. 3a). Additionally, we identified one or more type III secreted effectors of EHEC[34] (*espL*, *espR*, *espX*, *espY*) among 91% (269/294) of the GEMS non-DEC fecal isolates (Supplementary Data Set 2). Notably, one or more of these non-LEE effectors were identified in >99% of the GEMS isolates in the cryptic clade and in each of the phylogroups (A, B1, C, D, E, F) with the exception of phylogroups G (67%, 2/3) and B2 (0%, 0/23)(Supplementary Data Set 1). However, none of the GEMS non-DEC fecal isolates analyzed had an apparent functional type III secretion system similar to the locus of enterocyte effacement (LEE) region of EPEC and EHEC[35], which is thought to be required for secretion of these effectors.

The majority of these virulence factors had similar distributions among the GEMS non-DEC fecal isolates from cases versus controls (Fig. 4a and Supplementary Fig. 4a); however, select genes associated with adherence and biofilm formation (AFA/Dr adhesin, *ehaD)* or iron acquisition (*iroN)*, were more prevalent among the isolates from cases ($p < 0.05$)(Supplementary Data Set 2). Discriminant analysis of principal components[36] demonstrated these virulence factors are associated with phylogroup rather than with diarrhea (case/control) or geographic location (continent or GEMS site) (Fig. 4b and Supplementary Fig. 4). Clustering analysis based on the presence or absence of the virulence factors further demonstrated combinations of virulence factors associated with the phylogroups (Supplementary Fig. 3). The Chu cluster, which is involved in iron uptake[27,37], was among the loci most contributing to the phylogroup differences as it was absent from all genomes of phylogroups A and B1 but present among 80 to 100% of genomes in B2 and the other phylogroups and cryptic clade

(Supplementary Data Set 1 and Supplementary Fig. 3). Also, type II secretion system T2SSα[38] was detected in all genomes of phylogroups A and B2, but was absent from all but four genomes of phylogroup B1 (Supplementary Data Set 1).

Many of these chromosomally-encoded virulence factors were broadly distributed among the GEMS non-DEC fecal isolates; however, select mobile element encoded virulence factors were more associated with certain geographic locations (Supplementary Data Set 2). The colonization factor CS23 and the serine protease autotransporter EatA, which are often located on ETEC plasmids, were more prevalent among the isolates from GEMS sites in Africa compared to sites in Asia (Fig. 4a and Supplementary Data Set 2). EatA facilitates ETEC toxin delivery by degrading mucus[39,40] and is often co-located on virulence plasmids with the gene encoding the heat-stable enterotoxin (ST) of ETEC[17]. However, 12% (34/294) of the GEMS non-DEC fecal isolates examined in this study, which by our inclusion criteria are lacking the ST of ETEC, did in fact contain an *eatA* gene (Supplementary Data Set 2). The *eatA*-containing isolates were from all GEMS sites, represent four of the most prevalent phylogroups (A, B1, C, D) as well as the cryptic clade, and were identified in similar numbers of cases and controls (Supplementary Fig. 5a). Additionally, only seven of the 34 (21%) *eatA*-containing isolates were identified in lineages that also contained ETEC (Supplementary Data Set 1). The genes encoding CS23, EatA, and the EAST1 toxin of ETEC and EAEC, respectively, were identified among 81% (13/16) of the cryptic I isolates, which included isolates from six of the GEMS sites (Supplementary Fig. 5a). An IncFII plasmid containing *eatA* was identified among 64% (22/34) of the *eatA*-containing isolates, including all cryptic isolates (Supplementary Fig. 5b and Supplementary Data Set 5). Also, an *eatA*-containing IncB/O/K/Z plasmid was identified among five GEMS non-DEC fecal isolates from three phylogroups (A, C, and D) (Supplementary Fig. 5c and Supplementary Data Set 5). These findings demonstrate that at least two different plasmids have contributed to the dissemination of the *eatA* gene among the GEMS non-DEC fecal isolates.

## Distribution of antibiotic resistance genes and plasmids

The *E. coli* pathotype isolates characterized previously from GEMS contain an abundance of acquired antibiotic resistance genes (ARGs)[16–18]. Similarly, we identified one or more acquired ARGs in 80% (234/294) of the GEMS non-DEC fecal isolates (Fig. 5a, Supplementary Fig. 3, and Supplementary Data Set 2). Notably, 31% (47/152) of the GEMS non-DEC fecal isolates from control samples did not have any acquired ARGs compared to only 9% (13/142) of the isolates from cases that had no ARGs ($p = 0.0000028$) (Fig. 5b and Supplementary Data Set 3). Unlike the phylogroup associations observed for the virulence factors, the ARGs were associated with geographic location (Supplementary Fig. 4), which is consistent with previous findings for GEMS atypical EPEC[18]. Overall, the ARGs exhibited similar distributions among the GEMS non-DEC fecal isolates from cases versus controls (Supplementary Data Set 2). However, several ARGs that confer resistance to aminoglycosides (*aph*(3")-Ib and *aph*(6)-Id), sulfonamides (*sul2*), tetracycline (*tet*(B)), and penicillin antibiotics (*bla*TEM-1) were more prevalent among the isolates from cases compared to controls ($p < 0.05$)(Supplementary Data Set 2 and Supplementary Fig. 6a). The *sul1* or *sul2* genes were co-occurring with *dfrA* variants, *bla*TEM variants,

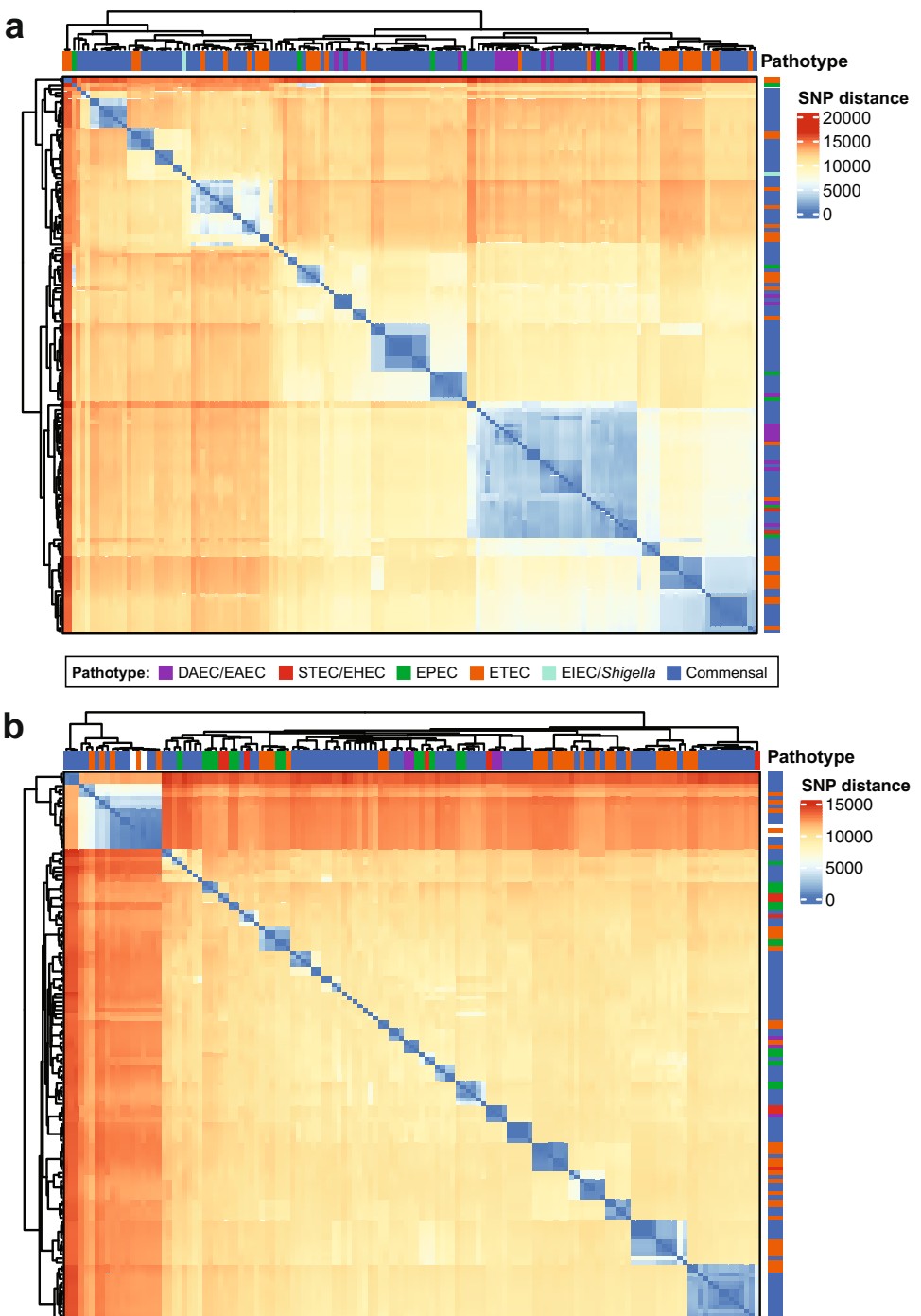

**Fig. 2 | Genomic similarity of the GEMS non-DEC fecal isolates to pathotype *E. coli*.** Heat maps of the pairwise SNP distances among genomes of phylogroup A (**a**) and phylogroups B1 and C (**b**). SNP distances were calculated for the SNPs analyzed in the phylogenomic analysis of Fig. 1. The row and column annotation of the heat maps indicates the pathotype association of each GEMS non-DEC or reference genome (see inset figure legend). The range of SNP distances and corresponding color scale is provided to the right of each heat map. The pairwise SNP distance matrix is available in Supplementary Data Set 3.

*tet(A)* or *tet(B)*, and *aph*(3")-Ib (*strA*) and/or *aph*(6)-Id (*strB*) in many of the GEMS non-DEC fecal isolates (Supplementary Fig. 6b). The *sul2*, *strA*, and *strB* genes were previously described as prevalent and co-occurring on plasmids among the atypical EPEC examined from GEMS[18].

ARGs of significant public health concern identified among the GEMS non-DEC fecal isolates include the extended-spectrum β-lactamase (ESBL) *bla*CTX-M-15, which was identified among 21% (27/126) of the isolates from Asia, but none of the isolates from Africa (Fig. 4a and Supplementary Data Set 2). In particular, 39% (16/41) of the isolates

from Pakistan contained *bla*CTX-M-15. Of the 27 *bla*CTX-M-15-containing GEMS non-DEC isolates, 30% (8/27) were identified as ST648 and ST131, which are *E. coli* lineages that have been previously linked to multidrug resistance and the dissemination of clinically important resistance determinants including ESBLs[41–43] (Supplementary Data Set 1). However, only one of the six (17%) ST131 compared to 78% (7/9) of the ST648 contained *bla*CTX-M-15. The other 19 GEMS non-DEC fecal isolates with the *bla*CTX-M-15 gene represent 16 different STs and were from phylogroups A, B1, B2, C, D, E, and F, demonstrating extensive genomic diversity among these fecal isolates that have acquired *bla*CTX-M-15.

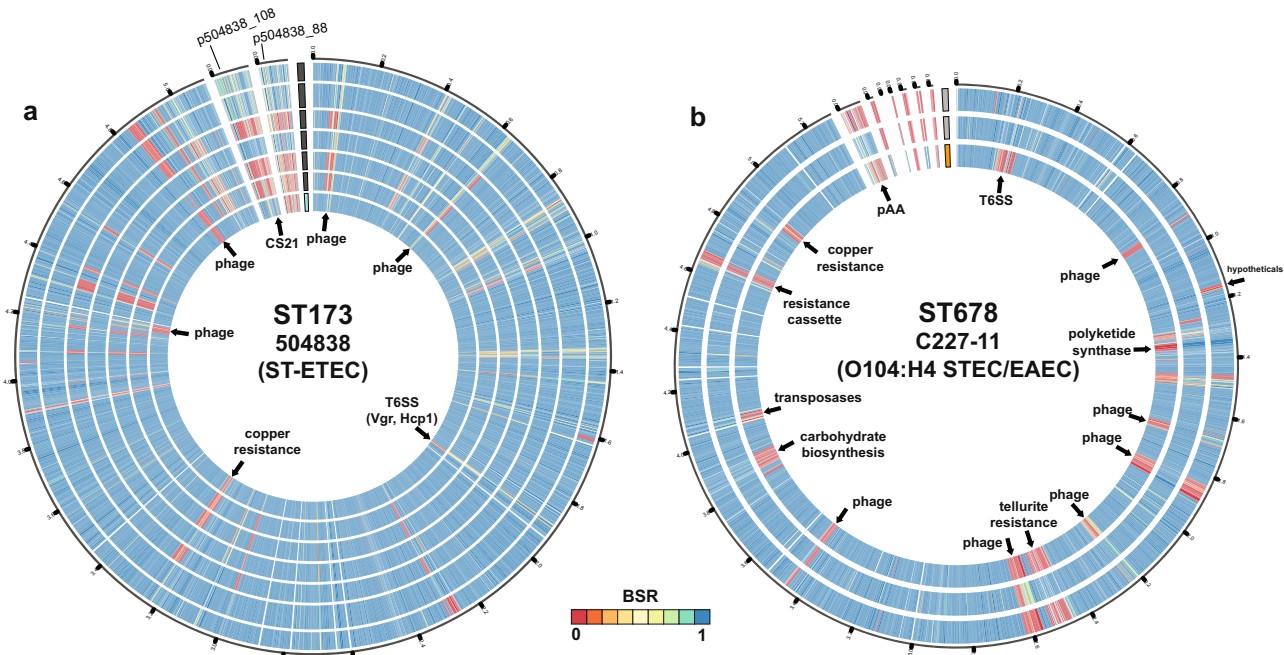

**Fig. 3 | Genomic similarity between GEMS non-DEC fecal isolates and previously described pathotype *E. coli* associated with severe diarrhea. a** Circular plot comparing the gene content of the previously described GEMS ST-ETEC isolate 504838[17] associated with a severe case of diarrhea to a GEMS non-DEC fecal isolate from a child with no diarrhea, and ETEC pathotype reference genomes that were all identified with as ST173. BSR values indicate the presence (blue), divergent similarity (yellow/green), and absence (red) of the 504838 genes among the other ST173 genomes. The largest segment of the plot is the chromosome and smaller segments are the two plasmids of ETEC isolate 504838 (p504838_108 and p504838_88). The label adjacent to each chromosome track indicates the type of genome: GEMS isolate from a control (light blue) and ETEC pathotype references (gray). **b** Circular plot comparing the gene content of the O104:H4 outbreak strain *E. coli* C227-11[25] to a GEMS non-DEC fecal isolate and other pathotype reference genomes of the ST678 lineage. The BSR values indicate the absence (red), divergent similarity (yellow/green), and presence (blue) of the C227-11 genes among the other genomes. The largest segment of the plot is the chromosome and smaller segments are each of the C227-11 plasmids. The label adjacent to each chromosome track indicates the type of genome: GEMS non-DEC fecal isolate from a case (orange) and two EAEC pathotype reference genomes (gray). BSR values that served as input for the plots are provided in Supplementary Data Set 4.

We also examined the prevalence of plasmid replicons among the GEMS non-DEC fecal isolates, as plasmids carry many of the ARGs described above, as well as *E. coli* virulence factors such as *eatA*[44,45]. One or more known plasmid replicons were identified in 89% (263/294) of the GEMS non-DEC isolates (Fig. 5c and Supplementary Data Set 2). Similar to the trend observed for the ARGs, there were a greater number of isolates from controls (15%, 23/152) compared to the isolates from cases (6%, 8/142) ($p = 0.012$) that contained none of the known plasmid replicons (Fig. 5d). Col-type plasmids, IncFII plasmids, and IncI plasmids were more prevalent among the GEMS non-DEC fecal isolates from cases compared to those from controls ($p = 0.003$, $p < 0.001$, $p = 0.001$, respectively) (Supplementary Data Set 2, Supplementary Fig. 3). Also, select IncFII plasmids were more prevalent among the GEMS non-DEC fecal isolates from Africa or Asia, as were IncI, IncX, and an IncB/O/K/Z plasmids ($p < 0.05$).

## Discussion

*E. coli* is one of the most studied bacterial species; however, the majority of *E. coli* genomic studies to date, including those investigating *E. coli* from GEMS[16–19], have focused on understanding diarrheagenic *E. coli*[1]. This focus on pathogens has largely ignored the *E. coli* that occupy the human gastrointestinal tract as non-diarrheagenic 'commensals'. A recent study described the genomic diversity among fecal *E. coli* from only healthy (non-diarrheal) children in the Gambia[22], which was one of the GEMS sites. Also, a limited number of previous studies have provided insight into the diversity and longitudinal dynamics of 'commensal' *E. coli* from human stools[20,46–49]; however, much remains unknown regarding the genomic diversity of non-diarrheagenic fecal *E. coli* from children, particularly from LMICs that suffer a considerable burden from diarrheal illness[15,50]. Here we describe extensive genomic diversity among the non-diarrheagenic fecal *E. coli* from children in south Asia and sub-Saharan Africa. This diversity is consistent with the limited prior studies that have described the genomic diversity of 'commensal' *E. coli* from children of these regions[5,21,22,49]. An important component of our study was the comparison of *E. coli* from children with and without diarrhea to determine whether isolates from certain lineages may be associated with gut health. Our findings demonstrate that overall, the GEMS non-DEC fecal isolates from children with and without diarrhea were identified in many of the same lineages. However, there were six lineages that contained exclusively GEMS non-DEC fecal isolates from children without diarrhea. These *E. coli* could be further investigated for a possible role in preventing the colonization and disease caused by pathotype *E. coli*[7,8] or other diarrheagenic species of bacteria[9].

An intriguing finding of this study is that nearly half of the GEMS non-DEC fecal isolates were identified in lineages that contained pathotype isolates, including characterized isolates linked to severe diarrheal illness. This number could be even greater following further investigation of these lineages after including additional pathotype isolates. There have been at least 15 genomic lineages of EPEC[16,51,52] and 21 lineages of ETEC[53] defined to date for which we have included representative genomes in our phylogenomic analysis. This large number of EPEC and ETEC lineages may have biased the similarity of the GEMS non-DEC *E. coli* in our study toward these pathotypes; however, there were few isolates identified in lineages containing EPEC references or other *E. coli* pathotypes compared to the ETEC lineages. This may in part be due to the fact that EPEC is defined by the presence of the chromosomally-encoded LEE region and the plasmid-encoded BFP in typical EPEC or lacking the plasmid in atypical EPEC, while ETEC is defined solely by the presence of the plasmid-encoded toxins, LT

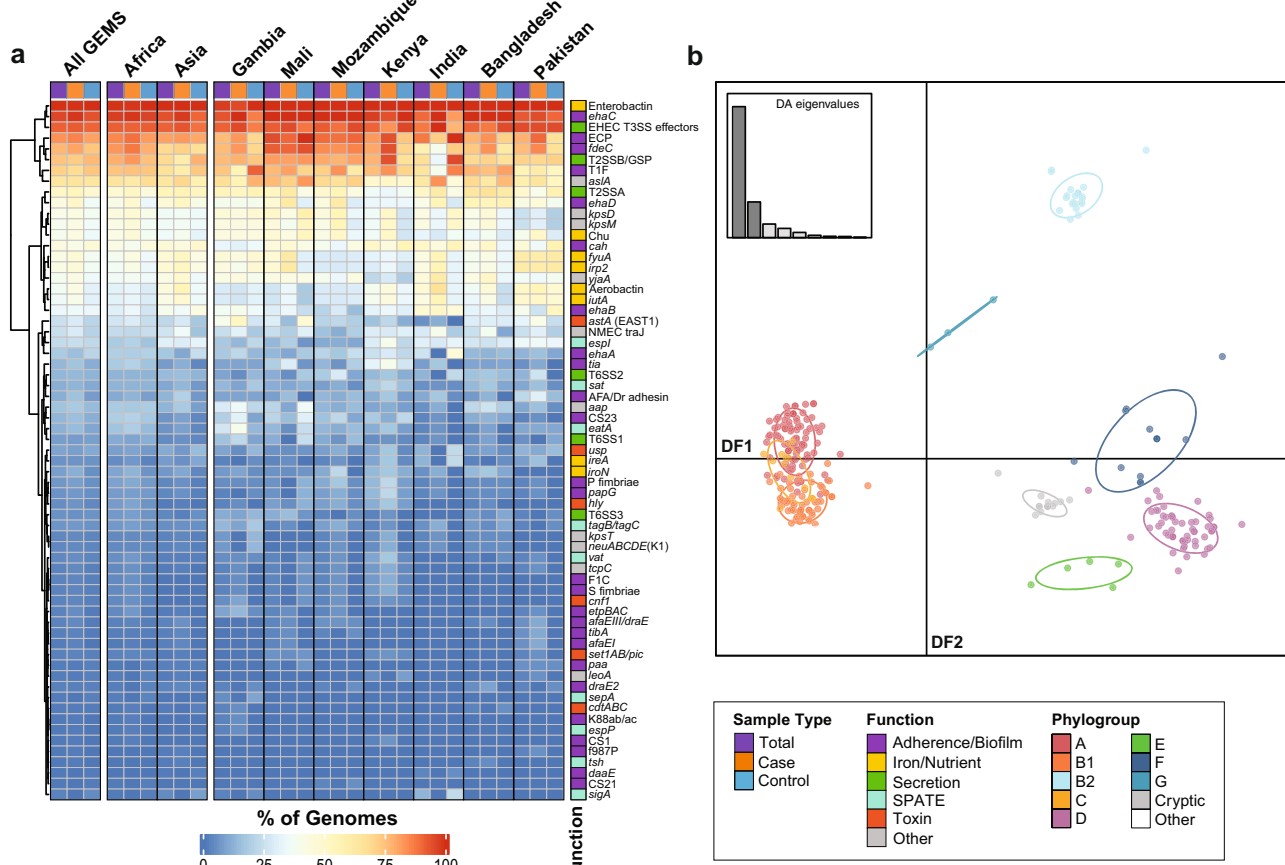

**Fig. 4 | Virulence factors identified among the GEMS non-DEC fecal isolates are associated with phylogroup. a** Heat map indicating the prevalence of virulence factors among the GEMS non-DEC genomes in total (purple) or associated with cases (orange) versus controls (blue). The distribution was examined across all sites combined (All GEMS), by continent (Africa vs. Asia), and by country. **b** Discriminant analysis of principal components (DAPC) was used to investigate the association of virulence factors with different phylogroups. The first 40 PCs and nine discriminant functions (DFs) were retained representing >96% of the conserved variance. An

inset bar plot shows the DA eigenvalues, which represent the ratio of the variance between groups over the ratio of the variance within groups for each of the nine DFs. The first two DFs shown as dark gray in the bar plot are represented in the scatter plot. Each point is an individual genome that is colored by phylogroup association and grouped with other genomes of the same phylogroup by an inertia ellipse. The virulence factors most contributing to DF1 and DF2 are *chu*, *espR1*, *espY3*, and *usp* for DF1, and *chu*, T2SSα, and *usp* for DF2. The input data for DAPC analysis is provided in the matrices of Supplementary Data Set 1.

and/or ST[1,2]. Further studies are necessary to determine whether certain lineages of *E. coli* can be exclusively linked to particular pathotypes, or whether the genetic diversity of pathotypes such as ETEC is a result of the host range of the ETEC virulence factor encoding plasmids.

The finding that non-diarrheagenic fecal *E. coli* can occupy the same lineage as pathotype *E. coli* also suggests that *E. coli* in these lineages may experience frequent loss and/or acquisition of the plasmid-encoded virulence factors required for pathotype designation. This would also presumably alter the type of interactions these *E. coli* would have with the host. In the current study, it was not possible to determine whether the GEMS non-DEC fecal isolates may have recently lost pathotype-defining virulence factors located on plasmids or other mobile elements as a result of laboratory passage or storage[44]. However, the selection of GEMS isolates for genome sequencing in our study was based on the preliminary PCR-based assessment performed at each of the GEMS sites upon initial culture[26], thus providing the most immediate snapshot of their pathotype associations, or lack thereof. We have previously noted a loss of virulence plasmids among EPEC from GEMS[16] and certain ETEC plasmids exhibit instability[54], which would impact the virulence potential as well as the classification of these isolates as belonging to a pathotype. Also, different growth conditions can influence the stability of *E. coli* virulence plasmids. The virulence plasmid of the EPEC archetype strain E2348/69 was stable in

the laboratory, but was rapidly lost during passage through human volunteers[55], suggesting factors in the host may impact the stability of plasmids that are central to virulence and pathotype classification. Although it is possible that some of the GEMS non-DEC fecal isolates may have lost pathotype-defining virulence plasmids during residence in vivo, we do not anticipate this is the case for all isolates. Thus, our findings highlight a gap in current knowledge of how readily *E. coli* may acquire, lose, and potentially re-acquire or remodel virulence plasmids, and how this may impact the ability of different *E. coli* to cause diarrhea. These findings also raise an important question about the dynamic nature of pathotypes and whether the long-held clinical paradigm of characterizing the dominant culturable pathotype *E. coli* isolate may be overlooking *E. coli* that had contributed to diarrhea as a pathotype isolate but lost the pathotype-defining virulence factors during the course of infection.

Another intriguing finding was that non-diarrheagenic fecal *E. coli* from the children with diarrhea more often carried acquired antibiotic resistance genes and plasmids compared to the *E. coli* from the children without diarrhea. Only three of the GEMS non-DEC fecal isolates, all from diarrhea, were from children that received an antibiotic prior to providing a GEMS stool sample, suggesting that antibiotic exposure likely did not select for the increased prevalence of ARGs among the children with diarrhea. However, it is not known whether the children with diarrhea may have received other antibiotic treatments for

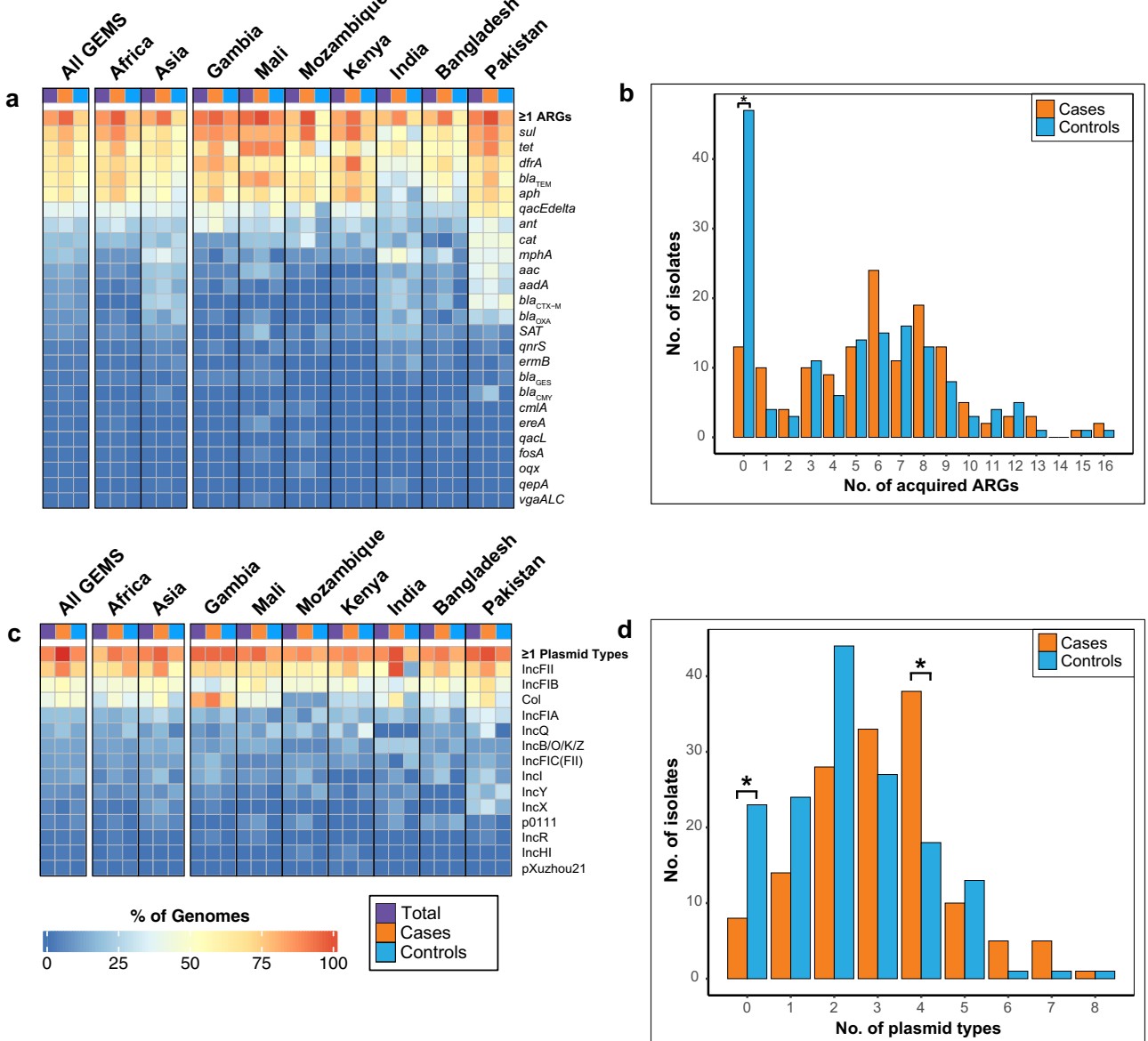

**Fig. 5 | Distribution and frequency of antibiotic resistance genes (ARGs) and plasmids among the GEMS non-DEC fecal isolates.** Heat maps indicating the prevalence of **a** acquired antibiotic resistance genes (ARGs) and **c** plasmids among all GEMS non-DEC fecal isolates in total (purple) or associated with cases (orange) versus controls (blue). The distribution was examined across all sites combined (All GEMS), by continent (Africa vs. Asia), and by country. The top row of each heat map indicates the percentage of genomes in each category that contain one or more of the ARGs or plasmid types identified. The numbers of **b** ARGs or **d** plasmid types identified per genome for each of the GEMS non-DEC fecal isolates. An asterisk denotes significant *p*-values by one-sided Fisher's exact test for the number of GEMS non-DEC fecal isolates from cases vs. controls that have no acquired ARGs (*p* = 0.0000028), no plasmids (*p* = 0.012), or four plasmids (*p* = 0.001). The ARG and plasmid counts are provided in Supplementary Data Set 2.

previous episodes of diarrhea or other illness in the months or years prior, which may have selected for an increased ARG prevalence. ETEC[17] and EPEC[16,18] from GEMS have been previously described with an abundance of acquired ARGs. Also, *E. coli* can acquire and maintain a multitude of different plasmid types and other mobile elements, including many that encode pathotype-specific virulence factors[44,45], and many of these plasmid types were prevalent among the GEMS isolates. This dynamic relationship with mobile elements has allowed *E. coli* to encode diverse virulence mechanisms and cause a wide variety of diseases[44,45,56]. Further studies are necessary to understand factors, host and microbial, that favor the acquisition of virulence or antibiotic resistance-conferring mobile elements by *E. coli*, and whether *E. coli* may further disseminate these mobile elements to other members of the microbiota.

In this work, we provide important insight into the largely unexplored genomic diversity of the non-diarrheagenic fecal *E. coli* from children in LMICs of south Asia and sub-Saharan Africa. Our study also greatly increases the number of 'commensal' *E. coli* genomes that have been described, which have been significantly understudied compared to pathotype *E. coli*. Further studies of the longitudinal dynamics of these *E. coli* could provide insight into whether they are transient or long-term colonizers[20,46], and whether they are able to confer a probiotic effect in preventing diarrhea. Also, it remains to be determined whether some of these *E. coli* isolates or lineages may have at one time possessed the mobile element-encoded virulence factors of diarrheagenic pathotypes and may be readily able to acquire these virulence factors, suggesting the pathogenic nature of *E. coli* may be more dynamic than previously appreciated.

## Methods

### GEMS study

The GEMS was a three-year case-control epidemiological investigation of the causes of diarrhea among children <5 years old in seven countries of south Asia and sub-Saharan Africa[14,15]. All details of the GEMS study design and microbiological methods for isolate identification have been previously described[14,15,26]. All subject data of GEMS enrollees was previously deposited in ClinEpiDb (https://clinepidb.org/ce/app) by the GEMS study coordinators. The clinical protocol was approved by ethics committees at the University of Maryland, Baltimore, MD, USA, and at every GEMS field site[15]. Written informed consent was obtained from the parent or primary caretaker of each participant before initiation of study activities[15]. This study received approval from The University of Maryland, Baltimore (UMB) Institutional Review Board protocols HP-00040030 and HP-00059433-7.

### Selection of non-diarrheagenic fecal *E. coli* from the GEMS isolate collection

At the outset of our study, we selected 50 different non-diarrheagenic fecal *E. coli*, each from a different GEMS enrollee from each of the seven geographic sites. These isolates were previously determined by PCR analysis to be lacking the characteristic virulence factors used to define the major diarrheagenic *E. coli* pathotypes (EPEC, STEC/EHEC, EAEC, and ETEC)[26]. The *E. coli* isolates were selected irrespective of whether there were other diarrheagenic pathogens or pathotype *E. coli* detected in the sample, thus representing a random selection of non-diarrheagenic fecal *E. coli* from both diarrheal and non-diarrheal samples of GEMS. No statistical method was used to predetermine sample size. Similar numbers of isolates were selected for cases and controls, and male and female study enrollees. Following the exclusion of isolates that did not grow or did not pass genome sequencing and assembly there were 336 non-DEC fecal isolates. Genomic analysis demonstrated that 42 of these isolates did contain characteristic virulence factors of the diarrheagenic pathotypes and these genomes were removed from further analysis as they did not fit our inclusion criteria for non-DEC fecal isolates. The 294 GEMS non-DEC genomes analyzed in this study represent 294 unique isolates from 294 different GEMS enrollees (Supplementary Data Set 1). These included 142 children exhibiting diarrhea at enrollment (cases) and 152 children that did not have diarrhea at enrollment or in the seven days prior to enrollment (controls). There were similar numbers of isolates from male (*n* = 160) and female (*n* = 134) children. The age range of the children was from one to 51 months with a median age of 14 months. There were similar numbers of isolates examined from each of the GEMS sites of south Asia (India, Bangladesh, and Pakistan) and sub-Saharan Africa (The Gambia, Mali, Mozambique, and Kenya).

We also examined the previously collected GEMS data[14,15] for the incidence of diarrheagenic pathogens co-occurring with the non-diarrheagenic fecal *E. coli*. Among the 142 cases were 132 (93%) that contained one or more diarrheagenic pathogens detected by GEMS (Supplementary Data Set 1). Among the most prevalent diarrheagenic pathogens identified were viruses (rotavirus, adenovirus, norovirus, saprovirus, and astrovirus)(89%, 263/294), parasites (*Entamoeba, Cryptosporidium*, and *Giardia*)(30%, 89/294), and other diarrheagenic bacteria (*Campylobacter* species, *Vibrio cholerae, Aeromonas* species, *Shigella, Salmonella* Typhi and non-Typhi)(23%, 67/294). There was a similar prevalence of most of the diarrheagenic pathogens among the case versus control samples, which was a finding of GEMS[15]. Only two of the samples (<1%) also contained a pathotype *E. coli* isolate (an EAEC and a typical EPEC). It is possible that multiple distinct *E. coli* were co-colonizing the children as has been previously described for children in the Gambia[22] and Tanzania[49] and the additional colonizing *E. coli* were not captured by the study design of GEMS, which saved one to three colonies of *E. coli* per stool[26]. In the current study we examined a single non-diarrheagenic fecal *E. coli* isolate from each child. Future studies would be necessary to understand the within-child population diversity of *E. coli*.

### Genome sequencing and assembly

The *E. coli* were grown overnight in Lysogeny Broth (LB)[57](Difco), and their genomic DNA was isolated using the GenElute Genomic DNA kit (Sigma-Aldrich). Genomic DNA was used to make paired-end sequencing libraries using the Kapa library kit, which were sequenced with 2 × 150 bp chemistry on the Ilumina HiSeq 4000 platform. Illumina reads were trimmed for quality using Trimmomatic v.0.36[58] and assembled using SPAdes v.3.11.1 with careful mismatch correction[59,60]. The assembly metrics and GenBank accession numbers of each assembly are provided in Supplementary Data Set 1.

### Multilocus sequence typing and serotypes

A multilocus sequence type (ST) was determined for each genome based on the scheme developed by Wirth et al. [61]. The sequences of each of the seven MLST loci (*adk, gyrB, fumC, icd, mdh, purA*, and *recA*) were identified in the genomes using BLASTN[62,63], and were used to obtain the allele numbers and sequence type of each genome using the BIGSdb[64] software. In silico serotype identification was performed on the assembled genomes using the sequence database v.1.0.0 of SerotypeFinder[65] and the default match criteria of >85% nucleotide identity and >60% alignment length.

### Phylogenomic analysis

The *E. coli* were compared by phylogenomic analysis using a single nucleotide polymorphism (SNP)-based approach[60,66]. The phylogenomic analysis in Fig. 1 included 294 GEMS *E. coli* genomes and 154 previously sequenced *E. coli* and *Shigella* reference genomes. Among the reference genomes were representatives of previously described *E. coli* phylogroups[67,68], as well as lineage-specific reference genomes for EHEC[51], EPEC[16,51,52], and ETEC[53] (Supplementary Data Set 1). Additional reference genomes of *E. albertii, E. fergusonii*, and the *Escherichia* cryptic clades[69] were included in Supplementary Fig. 1. While this reference genome collection cannot possibly capture every pathotype and 'commensal' lineage that exists, these reference genomes were selected to represent the defined pathotype-related genomic lineages and the overall diversity of the *E. coli* pathotypes and 'commensals' that have been previously described using genomics. The Northern Arizona SNP Pipeline (NASP v1.1.2)[70] was run with default parameters was used to identify core SNPs for each genome analyzed relative to the complete genome sequence of *E. coli* isolate IAI39 (NC_011750.1) as a reference. SNPs were filtered to remove sites in regions duplicated in the reference genome, sites with missing data, and monomorphic sites[70]. The remaining SNPs were concatenated for each genome representing a core SNP alignment for further analyses. There were 244,988 core SNPs identified among the genomes of Fig. 1, 301,042 core SNPs among the genomes in Supplementary Fig. 1, and 222,743 core SNPs among the *eatA*-containing genomes of Supplementary Fig. 5. The core SNP alignments were used as input to infer maximum-likelihood phylogenies with IQ-TREE v.1.6.12[71] using the GTR model of nucleotide substitution with GAMMA rate heterogeneity and ascertainment bias correction (GTR + G + ASC). Bootstrap support was determined using ultrafast bootstrap approximation (UFBoot2) run with 1,000 replicates and the bnni option to reduce overestimating support[72]. The phylogenies were midpoint rooted and labeled using the interactive tree of life (iTOL) v.5 software[73]. A bar plot of the phylogroup distributions was generated using ggplot2 v.3.3.5[74] in R v.4.1.0[75].

The *E. coli* that differed at one to two MLST loci (single or double locus variants, SLV or DLV) were grouped into lineages of closely related STs. The lineages also had significant bootstrap support (≥95) in the phylogenomic analysis. Three lineages contained an ST representing a triple locus variant (TLV) that was included based on

bootstrap support, core SNP differences, and patristic distances relative to other genomes in the lineage. The ST lineages are referred to by the most abundant ST in each lineage. Assigning these lineages based on ST provides important context for comparing the GEMS non-DEC fecal isolates to previously described *E. coli* lineages and for future isolate investigations, as STs remain an important classification and tracking tool in the clinical setting. The GEMS non-DEC genomes were further examined for core SNP differences. Pairwise SNP distances were determined for all genomes in the Fig. 1 phylogenomic analysis using snp-dists 0.8.2 (https://github.com/tseemann/snp-dists), and were visualized using the ComplexHeatmap v.2.8.0[76] package in R v.4.1.0[75]. The matrix of pairwise SNP distances is available in Supplementary Data Set 3. Pairwise patristic distances were inferred for all genomes in the Fig. 1 phylogenomic analysis using the cophenetic function of the ape 5.5 package[77] in R v.4.1.0[75,56].

### Identification of known plasmid types, antibiotic resistance genes, and virulence factors

Previously described plasmids represented by their incompatibility (Inc) type were identified in each of the GEMS *E. coli* genomes. Representative sequences of the Inc types of previously characterized plasmid families were detected in each of the GEMS genomes using BLASTN[62,63]. Plasmid types were considered present in a genome when the representative sequences from the PlasmidFinder database v.2021-11-29[78,79] were detected with the default criteria of ≥95% nucleotide identity and ≥60% alignment length.

Antibiotic resistance genes (ARGs) were identified in each genome using the resistance gene identifier (RGI) v.5.2.0 tool and the comprehensive antibiotic resistance database (CARD) v.3.1.4[80] with default parameters. The perfect and strict hits from RGI were further filtered to exclude the intrinsic ARGs and include only the ARGs that were likely acquired via mobile elements. The strict hits were also filtered to only include hits that had ≥90% identity and ≥80% alignment length of the reference to eliminate small fragments.

Previously described *E. coli* and *Shigella* virulence factors that are not pathotype-defining but provide an advantage during pathogenesis were detected in each of the *E. coli* genomes analyzed using BLAST score ratio (BSR) analysis with TBLASTN[16,17], with additional virulence genes included from the virulence factor database (VFDB)[27]. The genes identified with a BSR value ≥0.8 were considered present.

The distribution of antibiotic resistance genes, plasmids, and virulence factors was visualized as heat maps generated using the ComplexHeatmap v.2.8.0[76] package implemented in R v.4.1.0[75]. The rows and/or columns of the heat maps were clustered using the Euclidean distance metric and complete linkage method[76]. Bar plots of the numbers of ARGs or plasmids identified in each GEMS genome were generated using the ggplot2 v.3.3.5[74] package in R v.4.1.0[75].

### Comparison of GEMS non-DEC fecal isolates with pathotype *E. coli*

The predicted protein-coding genes of the previously characterized ST-ETEC 504838[17] were determined using an ergatis v.2-based prokaryotic annotation pipeline[81,82], while the protein-coding genes of the O104:H4 outbreak strain *E. coli* C227-11[25] were obtained from GenBank (GCA_000986765.1). Each gene was compared against the GEMS non-DEC genomes and additional reference *E. coli* in the same ST lineage using BLASTN BSR[17] (Supplementary Data Set 4). The BSR values for each genome were used to generate a circular plot of the gene presence or absence using Circos v.0.69-9[83].

### Identification of *eatA*-containing plasmids

The location of the *eatA* gene was determined in each of the GEMS non-DEC genome assemblies, and whether any of the *eatA*-containing contigs had a plasmid Inc type was determined using PlasmidFinder[78,79]. The plasmid contigs were annotated using RAST v.2.0[84]. The distribution of

the *eatA*-containing plasmids from GEMS non-DEC fecal isolates 100153 and 400621 were examined by detecting the protein-coding genes in each of the *eatA*-containing GEMS non-DEC fecal isolates by BLASTN BSR analysis[17] (Supplementary Data Set 5). The BSR values were visualized in heat maps generated using the ComplexHeatmap v.2.8.0[76] package implemented in R v.4.1.0[75]. The rows of the heat maps were clustered using the Euclidean distance metric and complete linkage method[76].

### Statistics & reproducibility

The association of the virulence factors and antibiotic resistance genes with the GEMS *E. coli* was examined by sample type (case or control), phylogroup, or geographic location (continent or GEMS site) by performing discriminant analysis of principal components (DAPC)[36] using the adegenet v. 2.1.4 package in R v.4.1.0[75]. The first 40 principal components were retained for the analysis of virulence factors, representing >96% of the conserved variance, and the first 30 PCs for the antibiotic resistance genes representing >97% of the conserved variance. The distribution of virulence factors, antibiotic resistance genes, and plasmids among GEMS *E. coli* by sample type (case or control) or geographic location (continent or country) was examined for statistical significance using a one-sided Fisher's exact test in R v.4.1.0[75]. Associations of phylogroups with sample type (case or control) or geographic location (continent or country) were also examined using Fisher's exact test.

### Reporting summary

Further information on research design is available in the Nature Portfolio Reporting Summary linked to this article.

## Data availability

All sequence data and genome assemblies generated in this study have been submitted to GenBank under the BioProject PRJNA611810. The individual assembly accession numbers and Illumina sequence read accession numbers are listed in Supplementary Data Set 1. Source data are provided with this paper.

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

## Acknowledgements

This project was funded in part by federal funds from the National Institute of Allergy and Infectious Diseases, National Institutes of Health, Department of Health and Human Services under grant number U19AI110820 (DAR). The original GEMS was funded by the Bill and Melinda Gates Foundation (grant number 38874). We thank the GEMS investigators, participants, and families for providing the bacterial isolates described in this study.

## Author contributions

T.H.H. and D.A.R. conceived and designed the study. S.M.T. provided isolates and J.M.M. extracted DNA. T.H.H. collected, analyzed and interpreted data. T.H.H. and D.A.R. wrote the manuscript and all authors edited for accuracy and clarity.

## Competing interests

The authors declare no competing interests.
