## [Peer Review File · Nature Communications]

Peer Review Comments file

Reviewer comments, first round review:

Reviewer #1 -Genomics of Escherichia coli pathotypes- (Remarks to the Author):

This manuscript represents a clear and concise description of a thorough genomic characterisation of almost 300 isolates of E. coli collected from children in developing countries together with a comparison of these isolates with reference InPEC genomes.

The originality of the study derives from the nature of the isolate collection. Most of the results recapitulate findings of previous studies. The authors have not cited a number of very relevant previous studies, most notably, the study by Touchon et al. PLoS Genet. 2020 Jun 12;16(6):e1008866.

Reviewer #2 -Diarrhoeagenic Escherichia coli- (Remarks to the Author):

The aim of this study was to examine the genomic diversity of E. coli from GEMS that did not belong to four of the known diarrheagenic E. coli (DEC) pathotypes, designated non-DEC, from children with and without diarrhea.

This is a very detailed and comprehensive study describing a high-quality analysis of analysis of a clinically relevant dataset.

This holistic approach to exploring the faecal microbiome is essential to understanding the disease process and the nature of pathogenicity of gastrointestinal pathogens. I have some thoughts and suggestions listed below. Essentially, I found the Results were difficult to read, but worth wading through to get to the Discussion, which I thoroughly enjoyed. This fascinating study makes an important contribution to the field.

Major points:

1. The most interesting findings of the study are related in the potential acquisition and loss of putative virulence factors and AMR determinants in the non-DEC in the study population. These findings should be included in the Abstract. Appreciate the word limit is limited but maybe need to edit re-focus to include the highlights articulated in the Discussion.
2. In the Introduction the authors should explain their rationale for analysing non-DEC isolates, and why they adopted the method of sequencing individual colonies rather than a metagenomic sequencing approach.
3. The Results section is overly long and difficult to read. Maybe it's the style of the journal but the first section (lines 95-117) is a description of the methods. Lines 118-127 describe the results from the GEMS study and perhaps would be better placed in the Introduction?
4. For clarity, the authors should clearly define each group of isolates (for example, (i) GEMS DEC, (ii) GEMS non-DEC and (iii) DEC and non-DEC reference genomes) and be consistent in using the designation throughout the manuscript.
5. The sections on ST lineages and genomic similarity are both very long and the take home messages are lost in the noise. These two sections would have more impact if the details were moved to supplementary data and the text focused on the key points.
6. The Discussion includes fascinating discourse on the nature of pathogenicity which is not captured in the Abstract. Again, appreciate it's not easy to include complex concepts in the Abstract but it is a key strength of this paper so would be good to try to include, if possible.

Minor points:

Abstract

Line 37 – The phrase “presumably non-diarrheagenic” is clumsy. It can be deleted in this sentence but suggest moving forward you define exactly what you mean and shorten to something easier to read, eg. “non-DEC”. Appreciate you want to leave open the possibility that members of this group

might have the potential to cause diarrhea but if up front you define the group as not belonging to the 4 pathotypes (EAEC, STEC, EPEC, ETEC), then you leave that door open while making the text easier to digest.

Lines 43-44 – I don't follow why non-DEC exhibiting genomic similarity to DEC suggests the pathogenic nature of E. coli may be more dynamic?

Lines 44-46 – Agree – but authors need to explain why studying non-DEC is important.

Introduction

Lines 48-49 – Suggest severe clinical outcomes rather than serious illness?

Line 53 – delete "presumable non-diarrheogenic"

Lines 55-56 – The role of non-DEC in preventing colonisation and disease is key to the rationale behind studying them so need to emphasise this point and return to it in the discussion.

Lines 59-61 – Are you referring to DEC or all pathotypes, including extraintestinal pathotypes?

Lines 67-69 – Need to explain why you didn't include EIEC?

Lines 78-83 – Suggest comparisons with other studies better placed in the Discussion??

Results

Lines 95-134 – No results in this section - mostly methods, results of the GEMs study and repeat of the aims of the study.

Line 55 – which E. coli? All E. coli in GEMs or just non-DEC?

Line 160 – How do the ST lineages equate to clonal complexes?

Line 161-173 – Suggest this level detail would be better in supplementary data?

Lines 190-249 – Suggest this section needs more focus and good edit

Lines 252- 349 – For me, the results described in this section are the most interesting and relevant, but they aren't mentioned in the abstract

Discussion

Line 345 – Are you referring to DEC or all pathotypes, including extraintestinal pathotypes?

Response to reviewers:

Reviewer #1 -Genomics of Escherichia coli pathotypes- (Remarks to the Author):

This manuscript represents a clear and concise description of a thorough genomic characterisation of almost 300 isolates of *E. coli* collected from children in developing countries together with a comparison of these isolates with reference InPEC genomes.

The originality of the study derives from the nature of the isolate collection. Most of the results recapitulate findings of previous studies. The authors have not cited a number of very relevant previous studies, most notably, the study by Touchon et al. PLoS Genet. 2020 Jun 12;16(6):e1008866.

Response: We thank the reviewer for the positive comments and have added this study as a reference.

Reviewer #2 -Diarrhoeagenic Escherichia coli- (Remarks to the Author):

The aim of this study was to examine the genomic diversity of *E. coli* from GEMS that did not belong to four of the known diarrheagenic *E. coli* (DEC) pathotypes, designated non-DEC, from children with and without diarrhea. This is a very detailed and comprehensive study describing a high-quality analysis of analysis of a clinically relevant dataset.

This holistic approach to exploring the faecal microbiome is essential to understanding the disease process and the nature of pathogenicity of gastrointestinal pathogens. I have some thoughts and suggestions listed below. Essentially, I found the Results were difficult to read, but worth wading through to get to the Discussion, which I thoroughly enjoyed. This fascinating study makes an important contribution to the field.

Response: We thank the reviewer for the positive comments.

Major points:

1. The most interesting findings of the study are related in the potential acquisition and loss of putative virulence factors and AMR determinants in the non-DEC in the study population. These findings should be included in the Abstract. Appreciate the word limit is limited but maybe need to edit re-focus to include the highlights articulated in the Discussion.

Response: We agree with the reviewer and have incorporated these findings in the abstract.

2. In the Introduction the authors should explain their rationale for analysing non-DEC isolates, and why they adopted the method of sequencing individual colonies rather than a metagenomic sequencing approach.

Response: We chose to examine cultured isolates in the current study as there is the potential that each child may have had multiple co-occurring *E. coli*. While strain-level analysis tools have improved significantly, the most definitive approach to link mobile elements such as plasmids to their host isolate is by examining cultured isolates. In future studies we do intend to look more into the population diversity of the *E. coli* as well as mobile elements and acquired resistance genes using metagenomics.

3. The Results section is overly long and difficult to read. Maybe it's the style of the journal but the first section (lines 95-117) is a description of the methods. Lines 118-127 describe the results from the GEMS study and perhaps would be better placed in the Introduction?

Response: We have moved this first section of results describing the strain selection to the methods to help shorten the results and avoid redundancy with the methods.

4. For clarity, the authors should clearly define each group of isolates (for example, (i) GEMS DEC, (ii) GEMS non-DEC and (iii) DEC and non-DEC reference genomes) and be consistent in using the designation throughout the manuscript.

Response: We have gone through the manuscript to clarify all descriptions of the different types of isolates. All GEMS isolates analyzed in this study are non-diarrheagenic fecal *E. coli* with the exception of one GEMS ETEC isolate we included from a prior published study for comparison. Whereas the reference genomes we refer to collectively as 'reference genomes' since they include a wide range of DEC and ExPEC as well as 'commensal' isolates.

5. The sections on ST lineages and genomic similarity are both very long and the take home messages are lost in the noise. These two sections would have more impact if the details were moved to supplementary data and the text focused on the key points.

Response: We have edited these sections to make them more focused on key points as suggested by the reviewer.

6. The Discussion includes fascinating discourse on the nature of pathogenicity which is not captured in the Abstract. Again, appreciate it's not easy to include complex concepts in the Abstract but it is a key strength of this paper so would be good to try to include, if possible.

Response: Thank you for this comment, we agree completely and have added a sentence to the abstract to highlight this concept.

Minor points:

Abstract

Line 37 – The phrase “presumably non-diarrheagenic” is clumsy. It can be deleted in this sentence but suggest moving forward you define exactly what you mean and shorten to something easier to read, eg. “non-DEC”. Appreciate you want to leave open the possibility that members of this group might have the potential to cause diarrhea but if up front you define the group as not belonging to the 4 pathotypes (EAEC, STEC, EPEC, ETEC), then you leave that door open while making the text easier to digest.

Response: We thank the reviewer for this suggestion and have revised the manuscript accordingly.

Lines 43-44 – I don't follow why non-DEC exhibiting genomic similarity to DEC suggests the pathogenic nature of *E. coli* may be more dynamic?

Response: We have clarified this statement by discussing the potential loss/acquisition of mobile elements that are characteristic of the diarrheagenic pathotypes.

Lines 44-46 – Agree – but authors need to explain why studying non-DEC is important.

Response: We thank the reviewer for this suggestion, and we have added a sentence to the abstract to establish this importance.

Introduction

Lines 48-49 – Suggest severe clinical outcomes rather than serious illness?

Response: Changed as suggested by the reviewer.

Line 53 – delete “presumable non-diarrheogenic”

Response: Changed as suggested by the reviewer.

Lines 55-56 – The role of non-DEC in preventing colonisation and disease is key to the rationale behind studying them so need to emphasise this point and return to it in the discussion.

Response: We appreciate this suggestion and have added this point to the abstract.

Lines 59-61 – Are you referring to DEC or all pathotypes, including extraintestinal pathotypes?

Response: We have clarified that this was diarrheogenic *E. coli* only.

Lines 67-69 – Need to explain why you didn't include EIEC?

Response: We have clarified that GEMS looked at only the four major DEC pathotypes, and later in the results specifically described that GEMS did not look at EIEC.

Lines 78-83 – Suggest comparisons with other studies better placed in the Discussion??

Response: We thank the reviewer for this suggestion. We have revised this paragraph to clarify several points being made and have moved several of the sentences to the discussion as suggested.

Results

Lines 95-134 – No results in this section - mostly methods, results of the GEMs study and repeat of the aims of the study.

Response: We have moved this section to the methods to cut down on the redundancy highlighted by the reviewer.

Line 55 – which *E. coli*? All *E. coli* in GEMs or just non-DEC?

Response: Clarified as suggested by the reviewer.

Line 160 – How do the ST lineages equate to clonal complexes?

Response: In most cases the ST lineages are analogous to the clonal complexes, but not in all cases. For example, CC10 encompasses a wide range of STs that in some instances differ by more than three loci. The ST (CC) predicted for each isolate by pubmlst.org is provided in Data Set S1 adjacent to the ST lineage referenced in this study. CC designations are also not available for all isolates, which is another reason why we opted to refer to isolates by their ST lineages.

Line 161-173 – Suggest this level detail would be better in supplementary data?

Response: We have shortened this section and focused the text to avoid repeating descriptions provided in the methods.

Lines 190-249 – Suggest this section needs more focus and good edit

Response: We have revised this section to focus on the overall pathotype similarities and have move the detailed examples comparing individual isolates to the supplementary data.

Lines 252- 349 – For me, the results described in this section are the most interesting and relevant, but they aren't mentioned in the abstract

Response: We thank the reviewer and have incorporated these findings in the abstract.

Discussion

Line 345 – Are you referring to DEC or all pathotypes, including extraintestinal pathotypes?

Response: Clarified as suggested by the reviewer.